# Surface Roughness and Color Stability of 3D-Printed Denture Base Materials after Simulated Brushing and Thermocycling

**DOI:** 10.3390/ma15186441

**Published:** 2022-09-16

**Authors:** Gülce Çakmak, Pedro Molinero-Mourelle, Marcella Silva De Paula, Canan Akay, Alfonso Rodriguez Cuellar, Mustafa Borga Donmez, Burak Yilmaz

**Affiliations:** 1Department of Reconstructive Dentistry and Gerodontology, School of Dental Medicine, University of Bern, 3012 Bern, Switzerland; 2Department of Prevention and Oral Rehabilitation, Universidade Federal de Goiás, Goiânia 74690-900, Goiás, Brazil; 3Department of Prosthodontics, Faculty of Dentistry, Eskişehir Osmangazi University, Eskisehir 26040, Turkey; 4Translational Medicine Research and Clinical Center, Eskişehir Osmangazi University, Eskisehir 26040, Turkey; 5Independent Researcher, Mexico City 03100, Mexico; 6Department of Prosthodontics, Faculty of Dentistry, Istinye University, Istanbul 34010, Turkey; 7Department of Restorative, Preventive, and Pediatric Dentistry, School of Dental Medicine, University of Bern, 3012 Bern, Switzerland; 8Division of Restorative and Prosthetic Dentistry, The Ohio State University, Columbus, OH 43210, USA

**Keywords:** 3D printing, color stability, denture base, surface roughness, thermocycling

## Abstract

Three-dimensional (3D) printing is increasingly used to fabricate denture base materials. However, information on the effect of simulated brushing and thermocycling on the surface roughness and color stability of 3D-printed denture base materials is lacking. The aim of this study was to evaluate the effect of brushing and thermocycling on the surface roughness and color stability of 3D-printed denture base materials and to compare with those of milled and heat-polymerized denture base resins. Disk-shaped specimens (Ø 10 mm × 2 mm) were prepared from 4 different denture base resins (NextDent Denture 3D+ (ND); Denturetec (SC); Polident d.o.o (PD); Promolux (CNV)) (n = 10). Surface roughness (R_a_) values were measured before and after polishing with a profilometer. Initial color coordinates were measured by using a spectrophotometer after polishing. Specimens were then consecutively subjected to simulated brushing (10,000 cycles), thermocycling (10,000 cycles), and brushing (10,000 cycles) again. R_a_ and color coordinates were measured after each interval. Color differences (ΔE_00_) between each interval were calculated and these values were further evaluated considering previously reported perceptibility (1.72 units) and acceptability (4.08 units) thresholds. Data were analyzed with Friedman, Kruskal–Wallis, and Mann–Whitney U tests (α = 0.05). R_a_ (*p* ≥ 0.051) and ΔE_00_ (*p* ≥ 0.061) values among different time intervals within each material were similar. Within each time interval, significant differences in R_a_ (*p* ≤ 0.002) and ΔE_00_ values (*p* ≤ 0.001) were observed among materials. Polishing, brushing, and thermocycling resulted in acceptable surface roughness for all materials that were either similar to or below 0.2 µm. Color of ND printed resin was affected by brushing and thermocycling. All materials had acceptable color stability when reported thresholds are considered.

## 1. Introduction

Edentulism is a common clinical condition that has been treated by using complete dentures for many years [1]. Polymethylmethacrylate (PMMA) has been the preferred material for the fabrication of complete dentures considering its low cost, polishability, ease of process, biocompatibility, and physical and optical properties [2,3,4,5]. Even though flask-pack-press manufacturing is still the most preferred technique for the fabrication of complete dentures [4], heat-polymerized PMMA was reported to have certain disadvantages such as rough surfaces and susceptibility to discoloration [3,4]. Therefore, milling and three-dimensional (3D) printing of denture bases by using materials with different chemical compositions have emerged as viable options [2,6].

Surface irregularities on a denture’s surface may lead to biofilm adherence and denture stomatitis [7], which makes denture cleaning a critical part of clinical success [8]. Various different denture cleaning methods have been recommended, yet brushing is considered to be the easiest and cheapest cleaning method [9,10]. However, previous studies have shown the significant effects of brushing on the surface roughness of PMMA [8,10,11,12,13,14], which is a known cause of plaque accumulation and biofilm formation [5,15]. A previous study has reported 0.2 µm as the clinically acceptable threshold value for surface roughness of dental materials [16]. In addition, surface roughness may impair the color stability of a complete denture [17]. Discoloration of a complete denture may indicate material damage and aging [2,18], which eventually can lead to the replacement of the denture [19,20].

An intraoral environment is a thermally dynamic medium due to temperature changes caused by consumed foods and beverages [21]. These temperature changes may result in thermal stress and subsequently degradation of the surface [22,23]. Previous studies have focused on surface roughness and color stability of 3D-printed denture base resins [1,4,7,14,21,24,25], while only two of those studies have investigated the effect of brushing of these parameters [7,14]. However, in both studies [7,14], 3D-printed denture base resin specimens were not compared with milled denture base resins and had a methodology that involved consecutive thermocycling, brushing, and staining. Therefore, the aim of the present study was to compare the surface roughness and color stability of 3D-printed denture base resins with those of milled and heat-polymerized PMMAs after brushing and thermocycling. The hypotheses were that (a) surface roughness of denture base resins would be affected by material type and time interval, and (b) color stability of denture base resins would be affected by material type and time interval.

## 2. Material and Methods

### 2.1. Specimen Preparation

Table 1 lists detailed information on the materials used in the present study. Four different denture base resins (NextDent Denture 3D+; NextDent B.V., Soesterberg, The Netherlands (ND); Denturetec; Saremco Dental AG, Rebstein, Switzerland (SC); Polident d.o.o; Polident, Volčja Draga, Slovenia (PD); Promolux; Merz Dental GmbH, Lütjenburg, Germany (CNV)) were used to fabricate 40 disk-shaped specimens (Ø10 mm × 2 mm), which was determined based on the results of previous studies (n = 10) [7,14,21,22]. For the fabrication of milled specimens (PD), a 10 mm-wide cylinder was designed in standard tessellation language (STL) format by using a design software (Meshmixer v3.5.474; Autodesk Inc, San Rafael, CA, USA). This design file was used to mill cylinders from prepolymerized PMMA disks (Milling unit M1; Zirkonzahn GmbH, Bruneck, Italy), which were then wet-sliced by using a precision cutter (Vari/cut VC-50; Leco Corporation, St Josephs, MI, USA) to obtain 2 mm-thick specimens. For the fabrication of 3D-printed specimens (ND and SC), a disk-shaped STL file with desired final dimensions was generated by using the same software. This STL file was transferred into nesting software (Composer v1.3.3; Asiga, Sydney, Australia for SC and RayWare; SprintRay Inc, Los Angeles, CA, USA for ND) and positioned with 60° angle to the build platform. After automatically generating supports, this configuration was duplicated 10 times and the specimens were printed with a layer thickness of 50 µm by using digital light processing (DLP) printers (MAX UV; Asiga, Sydney, Australia for SC and MoonRay S100; SprintRay Inc, Los Angeles, CA, USA for ND). For the fabrication of heat-polymerized specimens (CNV), which were considered as the control group, wax patterns with desired final dimensions were prepared and processed according to the traditional flask-press-pack technique (heat polymerization at 74 °C for 8 h) [4]. After fabrication, all specimens were smoothened by using #600 silicon carbide abrasive papers under running water. The final thickness (2 ±0.03 mm) of the specimens was controlled with a digital caliper (Model number NB60; Mitutoyo American Corporation, Providence, RI, USA) [2].

### 2.2. Baseline Surface Roughness and Color Coordinate Measurements

Three parallel linear traces, with a distance of 1 mm between them, were recorded. Perpendicular to those were another three parallel linear traces, again with a distance of 1 mm between them. There were recorded with a non-contact optical profilometer (FRT MicroProf 100, equipped with a CWL 300 µm sensor, resolution of 3 nm in z-dimension, Fries Research and Technology GmbH; Bergisch Gladbach, Germany) [26]. Each trace had a length of 5.5 mm and a pixel density of 5501 point/line. Baseline surface roughness (R_a_) of each trace was determined with the integrated software (Mark III, Fries Research & Technology GmbH; Bergisch Gladbach, Germany) according to the International Organization for Standardization 4287 standard [27] with cutoff values (Lc) of 0.8 mm, and the average of these traces were calculated. Then, all specimens were polished by using a slurry of pumice in water (Pumice fine; Benco Dental, Pittston, PA, USA) for 90 s (1500 rpm) [28] followed by a polishing paste (Fabulustre; Grobet USA, Carlstadt, NJ, USA) application for 90 s. After the specimens were ultrasonically cleaned (Eltrosonic Ultracleaner 07-08; Eltrosonic GmbH, Wiesbaden, Germany) in distilled water for 10 min at 40 kHz, R_a_ values were remeasured.

A digital spectrophotometer (CM-26d; Konica Minolta, Tokyo, Japan), which utilizes the Commission International de I’Eclairage (CIE) Standard (2-degree) human observer characteristics and CIE D65 illuminant, was used for the measurement of color coordinates. The same clinician (M.S.P.) performed all color measurements in a temperature- and humidity-controlled room with daylight over a gray background. A saturated sucrose solution was used for optical contact and the spectrophotometer was calibrated according to the manufacturer’s recommendation before each measurement. Three measurements were recorded for each specimen and these values were averaged.

### 2.3. Simulated Brushing and Thermocycling

After R_a_ and color measurements, specimens were subjected to simulated brushing (Bürstmaschine linear LR1; Syndicad Engineering, Munich, Deutschland) by using FDA-certified toothbrushes [29]. Total brushing time of 10,000 cycles (20,000 strokes, each cycle considered as a linear back and forth brushing action at a frequency of 1.5 Hz) was considered to replicate denture cleaning of approximately 2 years, as 10,000 strokes were reported to represent 1 year of denture cleaning [8,30,31]. In total, 6 brush heads were mounted to 6 separate slots and each brush applied a vertical load of 200 g directly onto the specimen surface. A soap slurry, which replicated the denture cleaning medium was homogenously prepared (T25 digital Ultra Turrax; IKA, Staufen, Germany) by mixing 1 part of alkali-free ground soap (Sibonet pH 6.5; Burnus GmbH, Darmstadt, Germany) and 3 parts of distilled water by weight [32]. Soap slurries were prepared and poured into each chamber of brushing machine until the surface of the specimens was covered. The toothbrushes and slurry were replaced with the new ones every 10,000 cycles for each specimen [33] and the test was performed at room temperature (23 °C). After brushing, the specimens were removed from the brushing machine, rinsed with distilled water, and gently air-dried.

The specimens were then subjected to 10,000 cycles of thermocycling (SD Mechatronik Thermocycler; SD Mechatronik GmbH, Westerham, Germany) at 5–55 °C in distilled water with a dwell time of 30 s and a transfer time of 10 s [14]. Finally, all specimens were subjected to an additional 10,000 brushing cycles as mentioned above. R_a_ and color coordinates were remeasured after each process. Color differences (ΔE_00_) between different time intervals were calculated by using the CIEDE2000 formula and the parametric factors (K_L_, K_C_, and K_H_) were set to 1 [34,35].

### 2.4. Statistical Analysis

Distribution of data was analyzed by using Kolmogorov–Smirnov tests. Due to non-normal distribution, non-parametric statistical analyses were performed. Friedman tests were used to analyze each material’s R_a_ and ΔE_00_ values within time intervals, while Bonferroni corrected Wilcoxon test was used to further evaluate the R_a_ data. Kruskal–Wallis and Mann–Whitney U tests were used to compare the R_a_ and ΔE_00_ values of the materials within each time interval. A statistical analysis software was used to perform all analyses (SPSS v25.0; IBM, Armonk, NY, USA) at a significance level of α = 0.05. Perceptibility and acceptability of ΔE_00_ values were further evaluated by the thresholds set by a previous study (perceptibility: 1.72 units, acceptability: 4.08 units) [19].

## 3. Results

Table 2 lists the descriptive statistics of R_a_ values. Even though Friedman tests showed significant differences among different timelines within each material (*p* < 0.001), Wilcoxon tests revealed that none of the pairwise comparisons led to a statistically significant difference (*p* ≥ 0.051). Kruskal–Wallis tests revealed significant differences among tested materials within each time interval (*p* ≤ 0.002). Before polish, ND had the highest (*p* ≤ 0.012) and PD had the lowest (*p* < 0.001) R_a_ values. In addition, SC had higher values than CNV (*p* = 0.006). SC had the lowest R_a_ after polish (*p* ≤ 0.012) and after thermocycling (*p* ≤ 0.018), whereas other materials had similar values (*p* ≥ 0.066). After first brushing cycle, ND had higher R_a_ values than PD (*p* < 0.001) and CNV (*p* = 0.024), while SC had similar values to those of other materials (*p* ≥ 0.054). After second brushing cycle, SC and PD had similar R_a_ values (*p* = 0.178) that were lower than that of CNV (*p* ≤ 0.012). However, ND had similar values to every other material tested (*p* ≥ 0.054).

None of the materials tested had significantly different ΔE_00_ values within different time intervals (*p* ≥ 0.061). However, Kruskal–Wallis tests showed that other than after first brushing cycle (*p* = 0.061), significant differences were observed among materials at different time intervals (*p* ≤ 0.001). ND had the highest ΔE_00_ values among materials after thermocycling (*p* ≤ 0.012) and after second brushing cycle (*p* ≤ 0.030), whereas the differences among other materials were nonsignificant in both time intervals (*p* ≥ 0.312). When overall (after polish-after second brushing cycle) ΔE_00_ were considered, SC and PD had similar values (*p* = 0.114) that were smaller than those of ND (*p* < 0.001) and CNV (*p* = 0.012). In addition, CNV had lower ΔE_00_ values than ND (*p* < 0.001) (Table 3). Figure 1, Figure 2 and Figure 3 show the differences in color coordinates of tested materials at each time interval.

## 4. Discussion

R_a_ values were significantly affected by the material type. Even though statistical analyses revealed that there were no significant differences among different time intervals within each material, the authors believe that there was a tendency towards lower R_a_ when before polish values were compared with those of other time intervals for most of the materials (Table 2). Given that the p-values of Wilcoxon tests were resulted from Bonferroni correction, which is a rather conservative analysis [36] it may be assumed that there were significant differences among different time intervals within each material. Therefore, first null hypothesis was accepted. Even though no significant differences were observed among ΔE_00_ values at different time intervals within materials, material type affected ΔE_00_ values within different time intervals. Thus, the second null hypothesis was also accepted.

None of the materials tested in the present study had R_a_ values similar to or lower than previously described clinical threshold value of 0.2 µm [16]. However, PD had significantly lower R_a_ than those of other materials before polish, which could be attributed to the fact that it was the only prepolymerized PMMA material tested. Millable PMMA pucks are fabricated under high pressure and high temperature, which leads to lower residual monomer content and higher degree of polymerization [22]. Nevertheless, after polish, all materials had lower R_a_ values than 0.2 µm, which is in line with previous studies [1,2,3,24,25]. Contrarily, Gad et al. [21] reported lower R_a_ values for 3D-printed resin when compared with heat-polymerized resin after polishing. The differences in polishing procedures and tested materials may have caused this contradiction. Nevertheless, consecutive procedures did not increase these values above 0.2 µm. Considering these findings, it can be speculated that tested denture base materials are resistant to surface deterioration after long-term use considering that brushing cycles in total represent approximately 4 years [8,30,31] and thermocycling simulate approximately one-year intraoral situation [37]. However, these findings should be substantiated with future studies that investigate the biofilm retention and bacterial plaque accumulation of tested denture base materials after simulated brushing and thermocycling.

A previous study has compared the R_a_ of 3D-printed and milled denture base materials after thermocycling, brushing, and staining [14]. The results of the present study partially agree with those of Alfouzan et al. [14] as they concluded that differences among tested materials after all procedures completed were nonsignificant and repetition of the procedures did not affect the R_a_. However, the authors [14] have also showed that brushing and staining increased the R_a_ values. The differences in tested materials and test design may be associated with contradicting results in present and Alfouzan et al.’s [14] studies. Nevertheless, R_a_ of 3D-printed denture base materials has not been investigated thoroughly. Therefore, future studies investigating the effect of simulated brushing and thermocycling on different mechanical properties with varying parameters would broaden the knowledge on limitations of these materials.

Effect of brushing on the R_a_ of denture base materials has been evaluated in previous studies, yet, contradicting results have been reported [8,9,10,11,12,13,14,38]. Similar to the present study, de Freitas Pontes et al. [11] and Lira et al. [38] showed that brushing did not affect R_a_, whereas other studies [8,9,10,12,13] found that brushing increased the R_a_. Brushing was performed by using a medium of soap slurry in the present study, while those studies [8,9,10,12,13] used dentifrice, which may have led to this difference. Even though a higher number of brushing cycles was applied, Shinawi’s [12] study may support this interpretation as they have tested the R_a_ of PD after brushing with dentifrice and reported increased values. Considering that no study has compared soap slurry and dentifrices on their effect on the surface properties of denture base materials, future studies are needed to substantiate the effect of brushing medium on denture base materials.

When ΔE_00_ values of each material-time interval pair were concerned, it was observed that only ND had ΔE_00_ values that were above the clinically perceptible threshold of 1.72 units [19] after thermocycling (ΔE_00_:2.1) and after all procedures were completed (ΔE_00_:2.83). Even though 3D-printed resin specimens were fabricated by using their respective proprietary DLP-based printers, compositions of the tested materials differ from each other. ND predominantly consists of ethoxylated bisphenol A dimethacrylate (Bis-EMA), which comprises more than 75%wt of the resin mixture. However, this ratio is between 25–50%wt in SC. In addition, titanium dioxide, silicon dioxide, phosphine oxide, and methacrylate monomers are also present in ND [39], whereas aliphatic urethane dimethacrylate and triethylene glycol dimethacrylate are present in SC [40]. These differences in material composition may have led to color stability differences observed between the two 3D-printed resins. Nevertheless, it can be stated that all materials had acceptable color stability even after all procedures were completed.

Color stability of 3D-printed denture base resins after brushing has been scarcely investigated and to the authors’ knowledge, only one study [7] has focused on this aspect. In their study, Alfouzan et al. [7] compared the color stability of two 3D-printed and one heat-polymerized denture base resins after consecutive thermocycling, brushing, and coffee immersion. The authors [7] showed that heat-polymerized denture base resin had the lowest color stability, while ND had lower color stability than the other 3D-printed resin tested. These results partially agree with those of the present study. The methodology of the present study may be associated with this difference, as the specimens were subjected to consecutive brushing, thermocycling, and brushing. Maintenance of a complete denture starts from the day it is delivered and continue throughout the use of the denture. Therefore, the authors believe that subjecting the specimens to simulated brushing before and after thermocycling is clinically relevant and simulates a realistic scenario to evaluate extrinsic and intrinsic color stability of denture base materials.

Lightness of SC and CNV remained similar throughout the procedures, while ND and PD had increased L* values after first brushing cycle. Thermocycling increased the L* values of ND, whereas those of PD was reduced. However, a second brushing cycle reduced the lightness of both materials. When redness of the materials was considered, SC increased while other materials decreased throughout the procedures. Yellowness of SC and PD increased after first brushing cycle, whereas CNV and ND had lower b* values. Thermocycling decreased the yellowness of all materials, while second brushing cycle did not affect the b* values evidently.

Significant differences were observed among the materials and different time intervals tested in the present study. The number of specimens was based on previous studies [7,14,21,22] rather than a power analysis, which may be a limitation. The present study aimed to evaluate the color stability of tested denture base resins after various treatments starting from after polish, given that unpolished surfaces are not suitable intraorally. However, color coordinates of the specimens were not measured before polish; thus, the effect of polishing on the color stability of tested denture base resins could not be evaluated. Another limitation of the present study was that tested denture base resins were limited to certain brands. In addition, the specimens were prepared to be compatible with the spectrophotometer and the brushing machine used. Thus, no international standards were followed for specimen dimensions. Previous studies have shown the effect of dentifrices on surface roughness [9,10,13]; thus, the design of the brushing process that comprised of a soap slurry as well as only one type of toothbrush was a limitation. In addition, no staining liquid was used in the present study. However, previous studies [4,7,18] have shown significant effect of colorants on the color stability of denture base materials. Finally, only two parameters were evaluated in the present study. However, other properties such as surface hardness, water sorption, translucency, and flexural strength are also effective on a denture base material’s clinical longevity. Future in vivo studies that involve diverse parameters such longer brushing durations, different brushing media, different staining solutions, and different chemical disinfectants are needed to comprehensively understand the limitations of the 3D-printed denture base materials tested.

## 5. Conclusions

Clinicians may consider tested denture base materials for the fabrication of removable prostheses, considering that polishing resulted in acceptable surface roughness for all tested materials, while brushing and thermocycling did not result in an increase above 0.2 µm. Even though the color stability of ND was affected by brushing and thermocycling and it had perceptible color change after thermocycling and after all procedures were completed, all materials had acceptable color stability when reported thresholds are considered.

## Figures and Tables

**Figure 1 materials-15-06441-f001:**
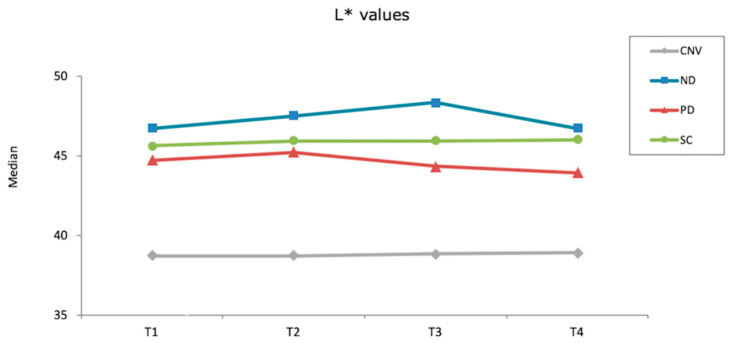
L* values of each material after each procedure (T1: After polish; T2: After first brushing cycle; T3: After thermocycling; T4: After second brushing cycle).

**Figure 2 materials-15-06441-f002:**
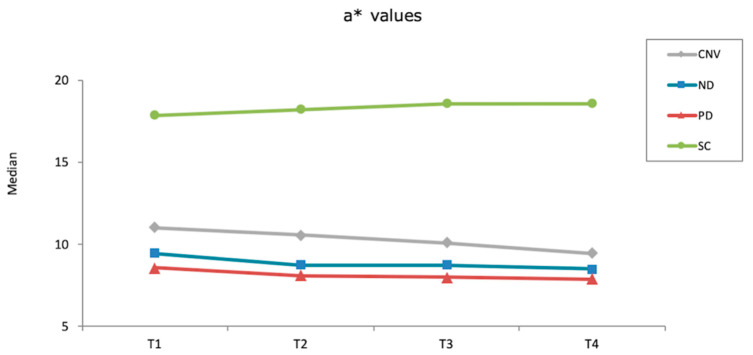
a* values of each material after each procedure (T1: After polish; T2: After first brushing cycle; T3: After thermocycling; T4: After second brushing cycle).

**Figure 3 materials-15-06441-f003:**
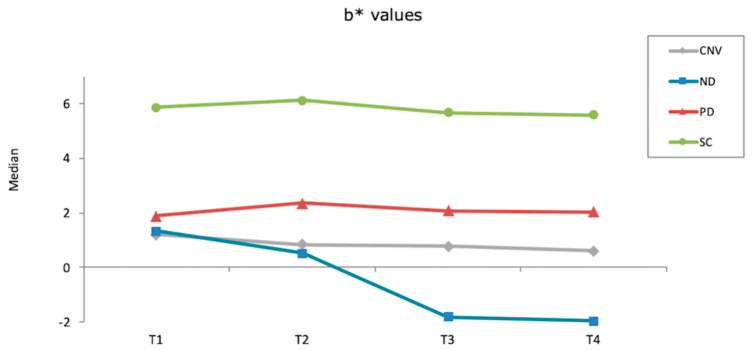
b* values of each material after each procedure (T1: After polish; T2: After first brushing cycle; T3: After thermocycling; T4: After second brushing cycle).

**Table 1 materials-15-06441-t001:** Materials used in this study.

Material	Type	Abbreviation	Manufacturer
NextDent Denture 3D+	3D-printed resin	ND	NextDent B.V., Soesterberg, The Netherlands
Denturetec	3D-printed resin	SC	Saremco Dental AG, Rebstein, Switzerland
Polident d.o.o	Prepolymerized PMMA disk	PD	Polident, Volčja Draga, Slovenia
Promolux	Heat-polymerized acrylic resin	CNV	Merz Dental GmbH, Lütjenburg, Germany

**Table 2 materials-15-06441-t002:** Median and 95% confidence interval R_a_ values (µm) of each material at different time intervals.

Material	Before Polish	After Polish	After First Brushing Cycle	After Thermocycling	After Second Brushing Cycle
ND	7.95 ^d^ (7.46–8.50)	0.13 ^b^ (0.13–0.14)	0.16 ^b^ (0.15–0.18)	0.15 ^b^ (0.14–0.16)	0.14 ^ab^ (0.14–0.17)
SC	4.18 ^c^ (3.57–4.83)	0.05 ^a^ (0.04–0.06)	0.07 ^ab^ (0.06–0.11)	0.06 ^a^ (0.05–0.06)	0.06 ^a^ (0.05–0.09)
PD	0.24 ^a^ (0.23–0.31)	0.14 ^b^ (0.08–0.22)	0.08 ^a^ (0.08–0.1)	0.10 ^b^ (0.09–0.13)	0.10 ^a^ (0.08–0.12)
CNV	1.05 ^b^ (0.69–2.81)	0.11 ^b^ (0.10–0.40)	0.12 ^a^ (0.10–0.14)	0.13 ^b^ (0.11–0.17)	0.20 ^b^ (0.15–0.22)

Different superscript letters indicate significant differences in columns (*p* < 0.05).

**Table 3 materials-15-06441-t003:** Median and 95% confidence interval ΔE_00_ values of each material between time intervals.

	Time Interval			
Material	After Polish- after First Brushing Cycle	After First Brushing Cycle- after Thermocycling	After Thermocycling- after Second Brushing Cycle	After Polish- after Second Brushing Cycle
ND	1.36 ^Aa^ (1.13–1.59)	2.10 ^Ab^ (1.83–2.43)	1.38 ^Ab^ (1.10–1.96)	2.83 ^Ac^ (2.57–2.93)
SC	0.57 ^Aa^ (0.28–0.87)	0.51 ^Aa^ (0.38–1.06)	0.38 ^Aa^ (0.17–0.47)	0.69 ^Aa^ (0.61–0.77)
PD	0.71 ^Aa^ (0.56–2.07)	0.83 ^Aa^ (0.45–1.71)	0.55 ^Aa^ (0.45–0.66)	0.90 ^Aa^ (0.77–1.38)
CNV	0.70 ^Aa^ (0.52–1.05)	0.68 ^Aa^ (0.52–0.78)	0.62 ^Aa^ (0.45–0.81)	1.62 ^Ab^ (1.28–1.91)

Different superscript letters indicate significant differences (uppercase letter for rows and lowercase letters for columns) (*p* < 0.05).

## Data Availability

The data presented in this study are available on request from the corresponding author.

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
