# Peer review of "Surface Roughness and Color Stability of 3D-Printed Denture Base Materials after Simulated Brushing and Thermocycling"

_materials, 2022, doi:10.3390/ma15186441_

Round 1

Reviewer 1 Report

Dear Authors,

Thank you very much for submitting your manuscript to Materials.

I fiind the topic very interesting especially in this digital era when there is a significant trend to replace conventional dental technological means with 3-D printing.

I hope that my remarks will help improve the quality of your paper.

Lines 52-53 – Please elaborate on the two studies that you mentioned.

Line 56 – Please replace „i” and „ii”with eather „a)” and „b” or „1)” and „2” as it can be misleading.

Line 60 – Specimen preparation – When you refer to certain material it is necessary to specify in „()” the producer, the registered mark sign, the producer, the city and the country. I would suggest to apply this formula at lines 61, 66, 71, 75 and so on throughout the entire manuscript.

Line 91 – Please mention the exact frequency of the ultrasonic cleaning device.

Line 127 – It is neccessary to mention why you applied the Krusckal-Wallis and Mann-Whitney tests.

Line 175 - Discussions section. Please elaborate on the biofilm retention.

Please refer in a comprehensive manner to the limitations of the study.

Line 268 – The Conclusions section should be rephrased in such a why that it reflects the cpractical impact of your findings in optimising future clinical protocols that involve 3D printing as a treatment solution of this type of patients.

Best regards!

Author Response

Dear Authors, 

Thank you very much for submitting your manuscript to Materials.

I find the topic very interesting especially in this digital era when there is a significant trend to replace conventional dental technological means with 3-D printing.

I hope that my remarks will help improve the quality of your paper.

Response: The authors would like to thank Reviewer #1 for their comments and contributions to the scientific quality of our paper.

Lines 52-53 – Please elaborate on the two studies that you mentioned.

Response: Previous studies on the color stability and surface roughness of 3D-printed denture base resins after brushing are now elaborated. The sentence reads “Previous studies have focused on surface roughness and color stability of 3D-printed denture base resins [1, 4, 7, 14, 21, 24, 25], while only 2 of those studies have investigated the effect of brushing of these parameters [7, 14]. However, in both studies [7, 14], 3D-printed denture base resin specimens were not compared with milled denture base resins and had a methodology that involved consecutive thermocycling, brushing, and staining.

Line 56 – Please replace „i” and „ii”with eather „a)” and „b” or „1)” and „2” as it can be misleading.

Response: Text is revised accordingly.

Line 60 – Specimen preparation – When you refer to certain material it is necessary to specify in „()” the producer, the registered mark sign, the producer, the city and the country. I would suggest to apply this formula at lines 61, 66, 71, 75 and so on throughout the entire manuscript.

Response: Entire manuscript is revised to elaborate the information on the materials used.

Line 91 – Please mention the exact frequency of the ultrasonic cleaning device.

Response: Exact frequency of the ultrasonic cleaner is given.

Line 127 – It is neccessary to mention why you applied the Krusckal-Wallis and Mann-Whitney tests.

Response: Both Ra and ΔE00 data were initially evaluated by using Kolmogoor-Smirnov tests. Due to non-normal distribution, non-parametric tests were used for further analyses. This is now elaborated in Statistical Analysis section.

Line 175 - Discussions section. Please elaborate on the biofilm retention.

Response: Even though the results of the present study showed that tested denture base resins were resistant to brushing and thermocycling considering the reported threshold of 0.2 µm, biofilm retention was not investigated in the present study and these findings should be supported with future studies involving this aspect. This is now highlighted in the second paragraph of Discussion, which reads “However, these findings should be substantiated with future studies that investigate the biofilm retention and bacterial plaque accumulation of tested denture base materials after simulated brushing and thermocycling.

Please refer in a comprehensive manner to the limitations of the study.

Response: Final paragraph of Discussion is elaborated with further limitations of the present study.

Line 268 – The Conclusions section should be rephrased in such a why that it reflects the practical impact of your findings in optimising future clinical protocols that involve 3D printing as a treatment solution of this type of patients.

Response: Conclusions section is revised and now reads “Clinicians may consider testeddenture base materials for the fabrication of removable prostheses, considering that polishing resulted in acceptable surface roughness for all materials, while brushing and thermocycling did not result in an increase above 0.2 µm. Even though the color stability of ND was affectedby brushing and thermocycling and it had perceptible color change after thermocycling and after all procedures were completed, all materials had acceptable color stability when reportedthresholds are considered.

Reviewer 2 Report

The manuscript presents some interesting results concerning surface roughness and colour stability of a variety of denture base materials. In my opinion, it may be interesting for a reader of Materials focussed on dental materials.

The following are my concerns and suggestions for improving the quality of the manuscript.

1) Surface roughness measurements.

In the Materials and methods section, line 85, the authors report that the profilometer has a resolution of 1 micrometer. This resolution is not sufficient for detecting surface roughness values close to or below the clinically acceptable threshold value of 0.2 micrometer (see Introduction section, line 43). Of course, it is unclear how roughness values reported in Table 1 have been determined. The authors need to check the instrument’s characteristics.

2) Materials and methods section. I suggest to report in a table the materials, the technologies and the acronyms in order to facilitate the reading and to distinguish the different materials. Moreover, it is unclear the meaning of the acronym SD (line 72 and line 75). Moreover, please use the correct symbol to denote the sexagesimal angle (line 72) and Celsius degree (line 78 and line 114).

3) Table 1.

- The unit of measure (micrometer) should be reported in the legend of table 1.

- The 95% confidence interval for the CNV After polish should be checked. Even if the distribution is not normal (it should be symmetric), it is quite unusual that the median value is so close to the lower extreme and it is so far from the upper extreme of the confidence interval.

- All the median values, as well as the confidence interval extremes values, should have the same number of significant decimals (i.e. two digits after the dot). For example, reported values of 0.1 should be written as 0.10.

- The statistics reported with superscript letters in the case of After First Brushing Cycle should be revised. It is very tough to believe that the median value of CNV (0.12) is significantly different from the ND value (0.16), while that of SC (0.07) is not significantly different from ND!

4) Table 2. In my opinion, this table confuses a reader. Why some P values are exactly reported while some others report >0.05?

In the text of the manuscript (Results section, lines 133-136) it is reported that: Wilcoxon tests revealed that none of the pairwise comparisons led to a statistically significant difference (P≥.051). In my opinion, this sentence is sufficient for describing the Wilcoxon test result without creating confusion. I suggest to delate table 2.

5) Table 3. All the median values, as well as the confidence interval extremes values, should have the same number of significant decimals (i.e. two digits after the dot).

6) Materials and methods section, Statistical Analysis subsection, lines 125-128. It is stated that a similar statistical approach is used to analyse Ra and DeltaE values. Instead, the Wilcoxon test has been performed only for Ra values!

7) Discussion section, p.8 line 234: …between 2 printed resin. Change with: …between the two 3D printed resins.

Author Response

The manuscript presents some interesting results concerning surface roughness and colour stability of a variety of denture base materials. In my opinion, it may be interesting for a reader of Materials focused on dental materials. The following are my concerns and suggestions for improving the quality of the manuscript.

Response: The authors would like to thank Reviewer #2 for their comments and contributions to the scientific quality of our paper.

1) Surface roughness measurements.

In the Materials and methods section, line 85, the authors report that the profilometer has a resolution of 1 micrometer. This resolution is not sufficient for detecting surface roughness values close to or below the clinically acceptable threshold value of 0.2 micrometer (see Introduction section, line 43). Of course, it is unclear how roughness values reported in Table 1 have been determined. The authors need to check the instrument’s characteristics.

Response: The authors would like to thank Reviewer #2 for pointing out this mistake. The text is revised as “Three parallel linear traces, with a distance of 1 mm between them, were recorded. Perpendicular to those, another three parallel linear traces, again with a distance of 1 mm between them, were recorded, with a non-contact optical profilometer (FRT MicroProf 100, equipped with a CWL 300 µm sensor, resolution of 3 nm in z-dimension, Fries Research & Technology GmbH; Bergisch Gladbach, Germany) [26]. Each trace had a length of 5.5 mm and a pixel density of 5501 point/line. Baseline surface roughness (Ra) of each trace was determined with the integrated software (Mark III, Fries Research & Technology GmbH; Bergisch Gladbach, Germany) according to the International Organization for Standardization 4287 standard [27] with cutoff values (Lc) of 0.8 mm, and the average of these traces were calculated.”

2) Materials and methods section. I suggest to report in a table the materials, the technologies and the acronyms in order to facilitate the reading and to distinguish the different materials. Moreover, it is unclear the meaning of the acronym SD (line 72 and line 75). Moreover, please use the correct symbol to denote the sexagesimal angle (line 72) and Celsius degree (line 78 and line 114).

Response: A new table (Table 1) is introduced and typos are corrected.

3) Table 1.

- The unit of measure (micrometer) should be reported in the legend of table 1.

Response: Legend of Table 2 (former Table 1) is revised.

- The 95% confidence interval for the CNV After polish should be checked. Even if the distribution is not normal (it should be symmetric), it is quite unusual that the median value is so close to the lower extreme and it is so far from the upper extreme of the confidence interval. 

Response: Statistical analyses of the present study were performed by a professional statistician, whom the authors have consulted again to ensure that the analyses and the results were represented correctly. Non-normal and asymmetric distribution of roughness values of CNV after polish can be seen in the graph below. The authors believe that this distribution also explains the asymmetric confidence interval.

- All the median values, as well as the confidence interval extremes values, should have the same number of significant decimals (i.e. two digits after the dot). For example, reported values of 0.1 should be written as 0.10.

Response: Former Table 1 (now Table 2) and former Table 3 are revised accordingly.

- The statistics reported with superscript letters in the case of After First Brushing Cycle should be revised. It is very tough to believe that the median value of CNV (0.12) is significantly different from the ND value (0.16), while that of SC (0.07) is not significantly different from ND!

Response: P-values of these comparisons were checked to ensure that no typos were made and they are both correct (p=0.024 and p=0.054 respectively). Differences can be attributed to the specific characteristics of non-symmetrical distributions.  

4) Table 2. In my opinion, this table confuses a reader. Why some P values are exactly reported while some others report >0.05?

Response: P values reported as “>.05” had an actual P value of “1”.

In the text of the manuscript (Results section, lines 133-136) it is reported that: Wilcoxon tests revealed that none of the pairwise comparisons led to a statistically significant difference (P≥.051). In my opinion, this sentence is sufficient for describing the Wilcoxon test result without creating confusion. I suggest to delate table 2.

Response: Table 2 is omitted as suggested.

5) Table 3. All the median values, as well as the confidence interval extremes values, should have the same number of significant decimals (i.e. two digits after the dot).

Response: Table 3 is revised accordingly.

6) Materials and methods section, Statistical Analysis subsection, lines 125-128. It is stated that a similar statistical approach is used to analyse Ra and DeltaE values. Instead, the Wilcoxon test has been performed only for Ra values!

Response: Wilcoxon test was not performed for DeltaE values as Friedman test revealed that the differences among different time intervals within each material were nonsignificant. The sentence is revised for clarity and now reads “Friedman tests were used to analyze each material’s Ra and ΔE00 values within time intervals, while Bonferroni corrected Wilcoxon test was used to further evaluate the Ra data.”

7) Discussion section, p.8 line 234: …between 2 printed resin. Change with: …between the two 3D printed resins.

Response: The sentence is revised accordingly.

Reviewer 3 Report

Article: Surface roughness and color stability of 3D-printed denture base materials after simulated brushing and thermocycling

Comments:

1.     It is recommended to increase the content of introduction. For example, the staining behavior or color stability of PMMA could be supplemented.

2.     In the Materials & Methods, what is the control group in this experiment?

3.     The information on four different denture base resins was insufficient. When making samples, were all four materials fabricated by milling, 3D-printed and heat-polymerized? Or did different processes use different materials separately? It is suggested that the author needs to strengthen the explanation.

4.     In the article, there is absolutely no comparison of the roughness and color stability of the specimens for the three processes (milled, 3D-printed, and heat-polymerized), which makes it difficult to understand why these three processes are specifically described in the experimental method.

5.     Why did the authors not measure the L*, a*, and b* of each material before the procedure (T0)?

6.     From lines 180 to 186, the authors are only based on the hypothesis of previous research, and the evidence is insufficient to reject the first null hypothesis. In addition, it is generally believed that different material types should have different properties. For example, why did the authors think that the color stability will not be affected by the material type in the second null hypothesis?

Author Response

Article: Surface roughness and color stability of 3D-printed denture base materials after simulated brushing and thermocycling

Comments:

1.     It is recommended to increase the content of introduction. For example, the staining behavior or color stability of PMMA could be supplemented.

Response: Shortcomings of heat-polymerized PMMA is now mentioned in the first paragraph of Introduction, which reads “Even though flask-pack-press manufacturing is still the most preferred technique for the fabrication of complete dentures [4], heat-polymerized PMMA was reported to have certain disadvantages such as rough surfaces and susceptibility to discoloration [3, 4]. Therefore, milling and 3-dimensional (3D) printing of denture bases by using materials with different chemical compositions have emerged as viable options [2, 6].”

2.     In the Materials & Methods, what is the control group in this experiment?

Response: Heat-polymerized PMMA (CNV) was considered as the control group of the present study. This is now elaborated in Materials and Method.

3.     The information on four different denture base resins was insufficient. When making samples, were all four materials fabricated by milling, 3D-printed and heat-polymerized? Or did different processes use different materials separately? It is suggested that the author needs to strengthen the explanation.

Response: A new table (Tabe 1) is introduced to elaborate the information on the denture base resins tested in the present study. First paragraph of Materials and Method section is also revised to clarify the fabrication methods used for each resin.

4.     In the article, there is absolutely no comparison of the roughness and color stability of the specimens for the three processes (milled, 3D-printed, and heat-polymerized), which makes it difficult to understand why these three processes are specifically described in the experimental method.

Response: The present study aimed to compare the surface roughness and color stability of 3D-printed denture base resins with those of milled and heat-polymerized PMMAs after brushing and thermocycling. However, in no part of the manuscript the authors have mentioned a comparison among different manufacturing methods. Even the hypotheses, which reads “The hypotheses were that a) surface roughness of denture base resins would be affected by material type and time interval, and b) color stability of denture base resins would be affected by material type and time interval are built based on material types rather than manufacturing methods. The authors believe a comparison among different manufacturing methods is not possible based on the results of the present study as only 4 denture base resins (2 additively, 1 subtractively, and 1 conventionally manufactured) were tested.

5.     Why did the authors not measure the L*, a*, and b* of each material before the procedure (T0)?

Response: Rather than their inherent color, the present study aimed to evaluate the color stability of tested denture base resins after various treatments starting from the condition that is suitable for delivery of a denture (after polish). This aspect is now elaborated in the final paragraph of Discussion. The section reads “The present study aimed to evaluate the color stability of tested denture base resins after various treatments starting from after polish, given that unpolished surfaces are not suitable intraorally. However, color coordinates of the specimens were not measured before polish; thus, the effect of polishing on the color stability of tested denture base resins could not be evaluated.

6.     From lines 180 to 186, the authors are only based on the hypothesis of previous research, and the evidence is insufficient to reject the first null hypothesis. In addition, it is generally believed that different material types should have different properties. For example, why did the authors think that the color stability will not be affected by the material type in the second null hypothesis?

Response: As suggested by Reviewer #3, tested denture base resins have different chemical compositions that could affect their material properties. Therefore, hypotheses of the present study are revised for clarity and integrity.
